# ENHANCING VISION TRANSFORMERS FOR OBJECT DETECTION VIA CONTEXT-AWARE TOKEN SELECTION AND PACKING

**Tianyi Zhang**
Department of Electrical and Computer Engineering
University of Minnesota
Minneapolis, MN 55455, USA
{zhan9167}@umn.edu

**Baoxin Li**
Department of Computer Science
University of Illinois Chicago
Chicago, IL 60607, USA
{baoxinli}@uic.edu

**Jae-sun Seo**
Cornell Tech
New York, NY 10044, USA
{js3528}@cornell.edu

**Yu Cao**
Department of Electrical and Computer Engineering
University of Minnesota
Minneapolis, MN 55455, USA
{yucao}@umn.edu

## ABSTRACT

In recent years, the long-range attention mechanism of vision transformers has driven significant performance breakthroughs across various computer vision tasks. However, these advancements come at the cost of inefficiency and substantial computational expense, especially when dealing with sparse data. While sparse attention mechanisms have been introduced to mitigate these issues by pruning tokens involved in attention, they often lack context-awareness and intelligence, frequently limiting the number of selected tokens uniformly across different inputs. To address these challenges, we propose a novel algorithm: Select and Pack Attention (SPA). SPA dynamically selects informative tokens using a low-cost gating layer and packs these selected tokens into new batches, allowing for a variable number of tokens to be used in GPU batch training and inference. Through extensive experiments on diverse datasets and multiple computer vision tasks, our method demonstrates superior performance and efficiency, including a 0.5-2.7 AP improvement in object detection and a 10.9%-24.9% reduction in computation.

## 1 INTRODUCTION

Recent advancements in computer vision tasks such as image classification, segmentation, and object detection have seen Vision Transformers (ViTs) surpass traditional convolutional approaches (Dosovitskiy et al., 2020; Liu et al., 2021; Xia et al., 2022; Chen et al., 2023) due to their powerful self-attention mechanisms. ViTs are particularly effective at capturing long-range dependencies, enabling the learning of global features that are crucial for complex visual understanding. However, this strength comes with a significant drawback: the computational overhead increases quadratically with the number of tokens (Liu et al., 2021; Hua et al., 2022), leading to excessive and often unnecessary computations among irrelevant tokens. This not only raises the computational burden but also risks degrading performance by incorporating extraneous, often redundant, information in typical computer vision tasks. As illustrated in Fig. 1, the self-attention mechanism in ViTs inadvertently processes a large amount of superfluous data (i.e. background tokens), exacerbating computational inefficiency and potentially degrading task performance by introducing irrelevant information into the model's

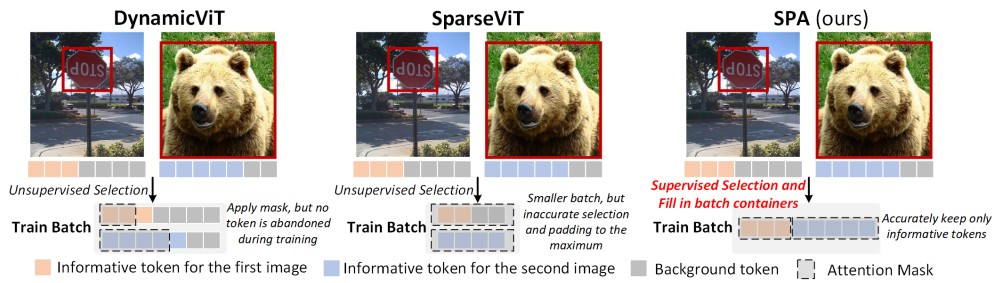

Figure 1: Previous sparse attention methods either reduce computation only during the inference stage or require padding the length of selected tokens to the maximum within a batch, which inevitably introduces background tokens. This leads to reduced efficiency and worse accuracy.

learning process. This issue is particularly severe when dealing with sparse data, such as in small object detection, where most pixels are not informative.

Numerous approaches have been proposed to address this issue by performing self-attention only on the most informative tokens. However, these methods still encounter significant challenges in both efficiency and performance.

- Efficiency: The constraints of GPU batch training, where images within a batch often contain non-uniform numbers of informative tokens, pose challenges to parallelizing computation effectively. Some methods, such as SparseViT (Chen et al., 2023), address this by padding all effective tokens to match the maximum number in the batch, leading to inefficiencies, as illustrated in Fig. 1. Other approaches, like DynamicViT (Rao et al., 2021) and EViT (Liang et al., 2022), reduce computation only during inference by discarding a fixed number of tokens. However, these methods still attend to all tokens during training, employing an attention mask to focus on informative tokens, which, along with the mask prediction module, can result in training costs that exceed those of a standard ViT. The Deformable Attention Transformer (DAT) (Xia et al., 2022), inspired by Deformable Convolutional Networks (DCN) (Dai et al., 2017), merely reduces the receptive field of query tokens while still computing all tokens, yielding minimal improvements in efficiency.

- Performance: Existing methods demonstrate effectiveness primarily in simpler tasks like image classification, where some degree of information loss is tolerable. However, their performance degrades in more complex tasks, such as object detection or instance segmentation, which demand richer semantic information. For example, DynamicSwin (Rao et al., 2023), a Swin-based DynamicViT, struggles in these scenarios due to inaccurate token selection, leading to significant information loss.

To address these challenges, we propose a novel Select and Pack Attention (SPA) mechanism that dynamically selects varying numbers of informative tokens from input batches, supervised by selection labels, and packs them into new batches for parallelized training. Specifically, we introduce a linear gating layer to generate scores for token selection, supervised by a multi-scale selection label derived from object labels (e.g., bounding boxes, instance segmentation labels). After selection, the chosen tokens are placed into uniform-sized package containers to form new batches, as illustrated in Fig. 1. For attention computation within each container, tokens attend only to those from the same original image, ignoring tokens from other images by using attention masks. Additionally, SPA can be effectively integrated with the window-based attention proposed by Swin Transformer (Liu et al., 2021), benefiting from the window shifting operation that captures information across windows. To prevent information loss across package containers, we shift the feature maps every two transformer blocks, ensuring that token pairs placed into containers vary, allowing the attention computation to encompass all tokens. Based on SPA, we propose a backbone network, Select and Pack Transformer (SPT), featuring a hierarchical architecture to generate image representations at various scales for downstream computer vision tasks. To avoid mis-selection at the early stage which may cause serious information loss (Xia et al., 2022), we leverage SPA from the third stage with the adapted image features. Ultimately, SPA addresses the efficiency issue by selecting only informative tokens and packing them into new batches, enabling efficient parallelized computation for both training and inference. Moreover, by leveraging selection label supervision, SPA improves

performance in complex computer vision tasks, such as object detection. Comprehensive experiments on three well-known datasets demonstrate the efficacy of SPA across multiple computer vision tasks.

To summarize, our main contributions are as follows:

- We propose a novel sparse attention mechanism, Select and Pack Attention (SPA), to enhance both the efficiency and performance of Vision Transformers. For efficiency, SPA dynamically selects informative tokens from images in a batch using a linear gating layer and packs them together to enable efficient GPU batch training and inference. For performance, we introduce a multi-scale selection label to explicitly supervise token selection, thereby outperforming existing methods even in complex computer vision tasks.

- By effectively integrating our SPA mechanism with Swin blocks, which use a window shifting trick to capture information across packages, we propose a backbone network with a hierarchical structure called the Select and Pack Transformer (SPT). SPT can generate features at various scales, making it suitable for many computer vision tasks.

- Through extensive experiments on diverse datasets, with lower computation cost, SPT outperforms state-of-the-art methods with a 0.5-2.7 AP improvement in object detection, a 1.3 AP gain in instance segmentation, and a 0.48 mAP increase in multi-label classification.

## 2 RELATED WORK

### 2.1 TRANSFORMER IN COMPUTER VISION

Given the remarkable success of transformers in natural language processing (NLP), this architectural paradigm is progressively permeating diverse computer vision tasks (Vaswani et al., 2017; Bao et al., 2021; Touvron et al., 2021; He et al., 2022; Zhang et al., 2024; Konstantinidis et al., 2023; Yang et al., 2022; Kang et al., 2022; Ni et al., 2024a;b; Zhou et al., 2024; Fan et al., 2024; Fan & Tao, 2024). For instance, Vision Transformer (ViT) divides input images into $16 \times 16$ patches, which are subsequently treated as tokens for the application of the attention mechanism (Dosovitskiy et al., 2020). In image segmentation, SAM (Kirillov et al., 2023) introduces a prompt-based algorithm, setting new benchmarks across state-of-the-art methods. For object detection, DETR conceptualizes it as a direct set prediction problem and designs a transformer-based network (Carion et al., 2020). DINO advances self-supervised learning to propose a novel network rooted in the ViT architecture (Caron et al., 2021). For super-resolution, transformers demonstrate exceptional capability in capturing long-range dependencies for improved visual representations (Liang et al., 2021; Zhang et al., 2023).

### 2.2 EFFICIENT TRANSFORMERS

Despite their advantages in global feature extraction via self-attention across all tokens, Vision Transformers (ViTs) are hindered by significant computational overhead because the computation of attention weights scales quadratically with the number of tokens. To address this challenge, efficient transformer variants (He et al., 2022; Child et al., 2019; Lingle, 2023) have explored multiple forms of sparsity to reduce attention cost. Swin Transformer (Liu et al., 2021) introduces window-based and shifted window-based self-attention mechanisms, significantly reducing computational demands within localized windows. DynamicViT (Rao et al., 2021) proposes a dynamic token sparsification framework to prune redundant tokens progressively and dynamically based on the input, SparseViT (Chen et al., 2023) optimizes computation by selecting tokens based on the $l_2$ norm of window activations, prioritizing features with higher scores. DAT (Xia et al., 2022) employs an offset network to refine the query token's receptive field, further enhancing computational efficiency. Fixed sparse patterns (Qiu et al., 2020; Zaheer et al., 2020) impose pre-defined attention masks to improve scalability for long sequences. Hybrid local–global attention mechanisms (Chu et al., 2021; Dong et al., 2022) restrict most computation to local windows while adding sparse global mixing tokens. Spatial factorization approaches (Ho et al., 2019) decompose 2D attention into row/column operations to reduce complexity. Learned sparsity frameworks (Wei et al., 2023) predict instance-dependent sparse attention patterns directly from features.

While effective, these approaches either (i) rely on heuristic or structure-driven sparsity that does not adapt well to downstream detection objectives, or (ii) lack dense supervision to ensure retention of small-object regions. In contrast, SPA learns token importance under explicit multi-scale supervision

from object-level labels. Moreover, our packaging mechanism addresses a practical limitation of dynamic sparse attention (irregular sequence lengths) by restoring batchwise parallelism.

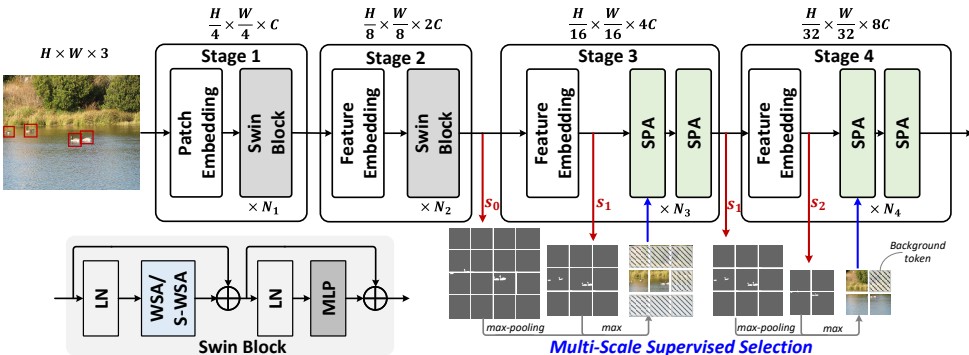

Figure 2: Overall architecture of Select and Packing Transformer (SPT). The hierarchical structure can generate features with various scales as common backbone networks. The SPA blocks in the last two stages can improve both efficiency and accuracy by disregarding uninformative tokens.

## 3 METHODOLOGY

### 3.1 OVERALL ARCHITECTURE OF SPT

As illustrated in Fig. 2, SPT features a hierarchical structure composed of four stages. Each stage generates image representations of varying sizes, resulting in a total of four different scales.

Specifically, suppose we consider a small $4 \times 4$ patch as a single token, the input image $\boldsymbol{x} \in \mathbb{R}^{H \times W \times 3}$ ($H$ and $W$ are the input image height and width), are progressively embedded into representations $\boldsymbol{r}_1 \in \mathbb{R}^{\frac{H}{4} \times \frac{W}{4} \times C}$, $\boldsymbol{r}_2 \in \mathbb{R}^{\frac{H}{8} \times \frac{W}{8} \times 2C}$, $\boldsymbol{r}_3 \in \mathbb{R}^{\frac{H}{16} \times \frac{W}{16} \times 4C}$, $\boldsymbol{r}_4 \in \mathbb{R}^{\frac{H}{32} \times \frac{W}{32} \times 8C}$ ($C$ is the embedding dimension of the first patch embedding layer) stage by stage. Each stage is structured around an embedding block for feature map downsampling, followed by $N_i$ transformer blocks tasked with feature learning ($N_i$ signifies the block count in the $i$th stage). Similar to Swin (Liu et al., 2021; 2022), the embedding block of the first stage $f_{\theta_1}$ employs a convolution layer while the subsequent embedding block consists of a patch merging layer that concatenates features as groups of $2 \times 2$ patches and a linear layer for feature projection. For the transformer blocks, the first two stages $f_{\theta_1}$, $f_{\theta_2}$ utilize standard Swin Transformer blocks, whereas the latter two stages $f_{\theta_3}$, $f_{\theta_4}$ incorporate our Select and Pack Attention (SPA) block. This design decision is informed by observations from DAT (Xia et al., 2022), which noted that early-stage transformer block replacement diminishes accuracy due to the model's inability to efficiently distinguish positive tokens based on shallow features. SPA blocks in the second and third stages not only generate outputs for subsequent layers but also transfer the score map to the next stage for the multi-scale supervision $\boldsymbol{s}_0 \in \mathbb{R}^{\frac{H}{8} \times \frac{W}{8} \times 1}$ and $\boldsymbol{s}_1 \in \mathbb{R}^{\frac{H}{16} \times \frac{W}{16} \times 1}$ for computing select loss. With the selection map $\boldsymbol{s}_2 \in \mathbb{R}^{\frac{H}{32} \times \frac{W}{32} \times 1}$ generated in the last stage, there are a total of three different scales. The complete process is as follows:

$$\boldsymbol{r}_1 = f_{\theta_1}(\boldsymbol{x}), \ \boldsymbol{r}_2 = f_{\theta_2}(\boldsymbol{r}_1), \ \boldsymbol{s}_0 = f_{\theta_g}(\boldsymbol{r}_2) \tag{1}$$

$$\boldsymbol{r}_3, \boldsymbol{s}_1 = f_{\theta_3}(\boldsymbol{r}_2, \boldsymbol{s}_0), \tag{2}$$

$$\boldsymbol{r}_4, \boldsymbol{s}_2 = f_{\theta_4}(\boldsymbol{r}_3, \boldsymbol{s}_1), \tag{3}$$

where $\boldsymbol{r}_1$, $\boldsymbol{r}_2$, $\boldsymbol{r}_3$, and $\boldsymbol{r}_4$ are output representations of four stages. $f_{\theta_1}$, $f_{\theta_2}$, $f_{\theta_3}$, and $f_{\theta_4}$ denote the four stage models. And $\boldsymbol{s}_0$, $\boldsymbol{s}_1$ and $\boldsymbol{s}_2$ denote the predicted score map for selection from the last three stages, separately. $f_{\theta_g}$ is the gating layer to generate scores for the output of stage 2.

In addition, given the higher efficiency of SPA blocks compared to Swin (Liu et al., 2021), we included 4 additional blocks at the third stage to enhance performance while maintaining a lower computation cost.

## 3.2 SELECT AND PACK ATTENTION (SPA)

Inspired by the gated networks in Mixture of Experts (MoE) (Petersen et al., 2022; Huang et al., 2020; Shazeer et al., 2017; Aoki et al., 2022; Chen et al., 2022) and heterogeneous federated learning (Lin et al., 2021; Ye et al., 2023), which adeptly guide models in selecting appropriate computational paths and enhancing task-specific generalization, we design a Select and Pack (SnP) block. This block utilizes a linear gating layer to select informative tokens (Detailed in Section A) and pack them into fixed-size package containers, generating new batches for GPU training or inference. While positive tokens undergo multi-head self-attention (MSA), negative tokens are directly passed to the feedforward network, as illustrated in Fig. 3a. SPA not only enhances the efficiency of the self-attention mechanism but also improves performance by focusing exclusively on informative tokens, effectively mitigating the potential for misleading context introduced by background tokens.

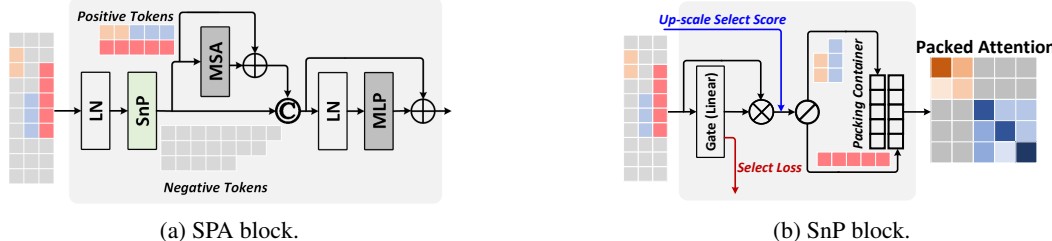

(a) SPA block.                    (b) SnP block.

Figure 3: (a) SPA computes attention (i.e., MSA) only for informative tokens. (b) SnP block selects informative tokens under multi-scale supervision and packs selected tokens for batch training and inference. The packed tokens attend to only tokens from the same image.

**Multi-Scale Supervised Selection.** Although token selection can be implicitly guided by the final objective, our experiments reveal that the gating layer tends to assign large values to all tokens, leading to the selection of too many tokens and reduced efficiency. To address this, we introduce a selection label based on object labels, which directly indicate areas of interest, such as instance segmentation masks or object detection bounding boxes. For segmentation, a binary mask assigns a value of 1 to all object pixels and 0 otherwise. For object detection, an aggregated binary mask is formed by stacking all bounding boxes. However, a single-scale label overly restricts token selection, causing significant information loss and poor performance. To mitigate this, we reduce the Gumbel-Softmax function's threshold and integrate multi-scale select labels. In Fig. 3b, each SPA block in SPT not only uses the selection scale matching the representation but also incorporates scores from up-scaled features, adjusted via max-pooling to match the correct feature size. This approach selects the maximum scores from two scales to include more informative tokens, thereby enhancing performance.

Specifically, given flattened input batch $r \in \mathbb{R}^{B \times N \times C}$ ($B$, $N$ and $C$ are the batch size, the length of each image representation and the number of channels, separately), the gate $f_{\theta_g}$ assigns scores $s \in \mathbb{R}^{B \times N \times 1}$ to each token. Then, we element-wise multiply the normalized scores by a sigmoid layer with the input representations to obtain the gated representations $r_g \in \mathbb{R}^{B \times N \times C}$. After that, we leverage the Gumbel-Softmax function (Jang et al., 2016) to separate positive tokens (i.e., informative tokens), $r_p \in \mathbb{R}^{N_p \times C}$ ($N_p$ is the number of positive tokens from all images in the batch) as follows:

$$s = \text{Max}(f_{\theta_g}(r), s_{up}), \tag{4}$$

$$r_g = \text{Sigmoid}(s) \odot r, \tag{5}$$

$$r_p = \text{Gumbel-Softmax}(s) \odot r_g \tag{6}$$

where $r$, $s_{up}$, $s$, $r_g$, $r_p$ denotes the input representation, scores for up-scale features, scores for this scale, gated representation, and output positive tokens, separately. And $\odot$ is element-wise multiplication with boardcasting. $f_{\theta_g}$ is the linear gating layer.

**Token Packing.** After the dynamic selection for each input image, the lengths of selected tokens vary. To avoid padding all tokens to the maximum length, which would introduce significant computational overhead, we pack the selected tokens into new batches. Inspired by (Dehghani et al., 2024), we set a series of package containers with a fixed length $L$ and fill them with the selected tokens. After packing all selected tokens, if the total number of tokens is not a multiple of the packing length, we only pad the last package. This approach is significantly more efficient than padding the selected

tokens for all images in the batch. Consequently, we obtain packed tokens, $\boldsymbol{p} \in \mathbb{R}^{B' \times L \times C}$ ($B'$ is the new batch size of packed tokens), and the number of tokens ($B' \times L$) is much smaller than the original input ($B \times N$), especially for sparse data. And the attention computation depends on $L$, similar to the window size $M$ of Swin. And we set $L$ to be $M^2$. Specifically, for input representation batch $\boldsymbol{r} \in \mathbb{R}^{B \times N \times C}$, the complexity of regular MSA, window-based self-attention (W-MSA), and SPA are as follows:

$$\Omega(\text{MSA}) = B(4NC^2 + 2N^2C), \tag{7}$$

$$\Omega(\text{W-MSA}) = B(4NC^2 + 2M^2NC), \tag{8}$$

$$\Omega(\text{SPA}) = B(NC + NC^2) + B'(3LC^2 + 2L^2C), \tag{9}$$

Compared to MSA, W-MSA is more efficient since the complexity is linear to the original token length $N$. However, our SPA is not only linear to $N$, the new batch size $B'$ is also much smaller than $B$, resulting in higher efficiency. For the self-attention of the packed tokens, we employ an attention mask to ensure that all tokens attend only to tokens from the same image, as illustrated in Fig. 3b.

Table 1: SPT achieves superior object detection performance on COCO2017 across three configurations while requiring lower overall computation. FLOPs are reported for the training stage. Additionally, we include the performance gain over the best sparse attention baseline. *For all base models, we adopted Cascade Mask RCNN framework for fair comparison.

| Methods | Attention | OD Performance | | | | | | FLOPs (G) | | #Params (M) | FPS (image/s) |
|---|---|---|---|---|---|---|---|---|---|---|---|
| | | AP | AP$_{50}$ | AP$_{75}$ | AP$_S$ | AP$_M$ | AP$_L$ | backbone | overall | | |
| Swin-T (Liu et al., 2021) | Dense | 46 | 68.1 | 50.3 | 31.2 | 49.2 | 60.1 | 96 | 267 | 48 | 50 |
| SparseViT (Chen et al., 2023) | Sparse | 42.4 | 63.3 | 46.4 | - | - | - | - | - | 48 | - |
| DAT-T (Xia et al., 2022) | Sparse | 44.4 | 67.6 | 48.5 | 28.3 | 47.5 | 58.5 | 101 | 272 | 48 | 46 |
| DynamicSwin-T (Rao et al., 2023) | Sparse | 44.3 | 65.9 | 48.5 | 30.2 | 47.1 | 58.5 | 101 | 272 | 48 | 46 |
| **SPT-T** (ours) | Sparse | **47.1 (+2.7)** | 68.9 | 51.6 | 31.7 | 50.6 | 60.9 | 90 (-10.9%) | 261 (-4.0%) | 55 | **54** |
| Swin-S (Liu et al., 2021) | Dense | 48.5 | 70.2 | 53.5 | 33.4 | 52.1 | 63.3 | 188 | 359 | 69 | 32 |
| DAT-S (Xia et al., 2022) | Sparse | 47.1 | 69.9 | 51.5 | 30.5 | 50.1 | 62.1 | 208 | 379 | 69 | 31 |
| DynamicSwin-S (Rao et al., 2023) | Sparse | 47.2 | 68.1 | 52.7 | 32.6 | 49.8 | 62.3 | 192 | 363 | 69 | 32 |
| **SPT-S** (ours) | Sparse | **49.3 (+2.1)** | 71 | 55.2 | 33.9 | 54.3 | 64.1 | 171 (-10.9%) | 342 (-5.8%) | 76 | **33** |
| Swin-B* Liu et al. (2021) | Dense | 51.9 | 70.5 | 56.4 | 35.4 | 55.2 | 67.4 | 332 | 982 | 145 | 11 |
| DynamicSwin-B (Rao et al., 2023) | Sparse | 50.5 | 69.7 | 58.9 | 34.7 | 53.2 | 66.1 | 341 | 991 | 145 | 11 |
| **SPT-B** (ours) | Sparse | **53.2 (+2.7)** | 71.3 | 58.9 | 36 | 57.6 | 67.9 | 294 (-11.4%) | 944 (-3.9%) | 153 | **12** |

### 3.3 Loss Function

The loss function of SPT comprises the loss for the target task and the selection loss. For the selection loss, we adopt binary cross-entropy and sum over all SPA blocks as follows,

$$\mathcal{L}_{select} = -\sum_{block} \left( \boldsymbol{y} \log \boldsymbol{s} + (1 - \boldsymbol{y}) \log(1 - \boldsymbol{s}) \right) \tag{10}$$

where $\boldsymbol{s}$ is the normalized score map through Sigmoid layer, and $\boldsymbol{y}$ denotes the ground truth label.

The overall loss function of SPT is $\mathcal{L}_{SPT} = \mathcal{L}_{task} + \alpha \mathcal{L}_{select}$, where $\alpha$ is a hyperparameter.

## 4 Experimental Results

### 4.1 Data and Experimental Setup

To evaluate the effectiveness of our SPT Transformer in complex computer vision tasks, we conducted extensive experiments on object detection using the widely adopted COCO2017 and BDD100K datasets. The COCO2017 dataset includes objects from 80 categories, while BDD100K comprises 10 categories. Beyond standard object detection, we further tested our method on the challenging task of early object detection, where objects are predominantly small, making both object detection and token selection significantly more difficult. This scenario underscores the importance of accurate token selection, as errors can result in the loss of entire objects, thereby highlighting the robustness of our approach. To simulate this scenario, we adopted the method proposed in (Zhang et al., 2024). Specifically, we computed the object-to-pixel ratios across entire images and selected samples from the BDD100K dataset with object ratios below 25%, creating the BDD-S dataset.

Additionally, to further showcase the robustness of our SPT Transformer, we extend our experiments to additional computer vision tasks, including instance segmentation on COCO2017 and multi-label

Table 2: SPT-based Mask RCNN achieves better object detection performance with less total computation on BDD100K for all three configurations. FLOPS is for training stage. We adopted the same image resolution as COCO2017.

| Methods | Attention | OD Performance | | | | | | FLOPs (G) | | #Params (M) | FPS (image/s) |
|---|---|---|---|---|---|---|---|---|---|---|---|
| | | AP | $AP_{50}$ | $AP_{75}$ | $AP_S$ | $AP_M$ | $AP_L$ | backbone | overall | | |
| Swin-T (Liu et al., 2021) | Dense | 22.4 | 34.6 | 24.6 | 7.9 | 20.1 | 46.8 | 96 | 267 | 48 | 50 |
| DynamicSwin-T (Rao et al., 2023) | Sparse | 22.0 | 33.1 | 23.7 | 9.2 | 19.7 | 44.7 | 101 | 272 | 48 | 46 |
| **SPT-T** (ours) | Sparse | **22.6 (+0.6)** | 33.1 | 24.6 | 8.8 | 18.5 | 47.7 | **84 (-16.8%)** | **255 (-6.3%)** | 55 | **58** |
| Swin-S (Liu et al., 2021) | Dense | 22.6 | 34.9 | 24.9 | 8.1 | 20.3 | 47.1 | 188 | 359 | 69 | 32 |
| DynamicSwin-S (Rao et al., 2023) | Sparse | 22.3 | 33.4 | 24.2 | 9.0 | 19.9 | 45.4 | 192 | 363 | 69 | 32 |
| **SPT-S** (ours) | Sparse | **22.9 (+0.6)** | 33.8 | 25.2 | 8.7 | 19.8 | 48.2 | **155 (-19.3%)** | **326 (-10.2%)** | 76 | **34** |
| Swin-B (Liu et al., 2021) | Dense | 22.7 | 35.1 | 25.1 | 8.2 | 20.5 | 47.6 | 332 | 508 | 107 | 18 |
| DynamicSwin-B (Rao et al., 2023) | Sparse | 22.4 | 33.6 | 24.6 | 8.5 | 20.2 | 46.3 | 341 | 517 | 107 | 18 |
| **SPT-B** (ours) | Sparse | **23.1 (+0.7)** | 34.3 | 25.5 | 8.2 | 20.6 | 48.6 | **256 (-24.9%)** | **432 (-16.4%)** | 115 | **20** |

Table 3: For early object detection on the more challenging BDD-S dataset, SPT also outperforms baseline models with 20.8-22.4% backbone computation cost reduction when compared to baselines.

| Methods | Attention | OD Performance | | | | | | FLOPs (G) | | #Params (M) | FPS (image/s) |
|---|---|---|---|---|---|---|---|---|---|---|---|
| | | AP | $AP_{50}$ | $AP_{75}$ | $AP_S$ | $AP_M$ | $AP_L$ | backbone | overall | | |
| Swin-T (Liu et al., 2021) | Dense | 5.5 | 8.6 | 5.9 | 1.4 | 2.7 | 15.4 | 96 | 267 | 48 | 50 |
| DynamicSwin-T (Rao et al., 2023) | Sparse | 4.7 | 8.4 | 3.7 | 1.0 | 2.6 | 12.8 | 101 | 272 | 48 | 46 |
| **SPT-T** (ours) | Sparse | **5.6 (+0.9)** | 9.0 | 6.4 | 1.5 | 2.7 | 15.4 | **80 (-20.8%)** | **251 (-7.7%)** | 55 | **62** |
| Swin-S (Liu et al., 2021) | Dense | 5.4 | 9.0 | 6.0 | 1.7 | 2.9 | 14.1 | 188 | 359 | 69 | 32 |
| DynamicSwin-S (Rao et al., 2023) | Sparse | 5.2 | 8.2 | 6.0 | 0.9 | 2.3 | 14.3 | 192 | 363 | 69 | 32 |
| **SPT-S** (ours) | Sparse | **5.7 (+0.5)** | 9.2 | 6.6 | 1.7 | 2.9 | 15.5 | **149 (-22.4%)** | **320 (-11.8%)** | 76 | **35** |

classification on PASCAL VOC 2012 dataset. For multi-label classification, we also select images with object pixel ratios smaller than 25% to show the effectiveness of SPT in challenging scenarios.

To evaluate object detection and instance segmentation, we utilize the Mask RCNN framework, and replace the backbone network with our SPT or other baselines. We adopt the default training settings, such as 36 max training epochs, batch size of 2. In addition, we set the threshold of Gumbel-Softmax to 0.01, and set the select loss weight $\alpha$ to 0.01. We set the window size of Swin block to 7, and the container length of SPA to 49. Experiments were performed on two Linux servers, each outfitted with dual NVIDIA L40S GPUs.

## 4.2 SPT FOR OBJECT DETECTION

In Table 1, we present a comparison of SPT with other baselines for the tiny, small, and base configurations (i.e., SPT-T, SPT-S, and SPT-B) on the COCO2017 dataset. For baselines, we focused on hierarchical backbones, as vanilla ViT-based methods, such as EViT (Liang et al., 2022), produce single-scale outputs and perform significantly worse on object detection tasks compared to hierarchical architectures like Swin Transformer (Liu et al., 2021; 2022), which provide feature pyramids. In addition to comparisons with sparse attention approaches, we included Swin results as a dense attention reference. Under the same settings, our approach outperforms all baselines including dense attention (i.e., Swin). Compared to the best-performing sparse attention baseline, SPT delivers AP improvements ranging from 2.1 to 2.7. Similarly, in Table 2, we report the results on the BDD100K dataset, where SPT surpasses all baselines and achieves state-of-the-art object detection performance, with an AP improvement of 0.6 to 0.7 over DynamicSwin (Rao et al., 2023). SPT-B attains the highest AP of 23.1.

**Early Object Detection Performance.** For the performance of early object detection on the BDD-S dataset, we observed that the base model underperforms compared to the tiny and small versions in this scenario. Consequently, we included only the results of the tiny and small models in Table 3. Notably, SPT-T and SPT-S achieved performance improvements of 19.1% and 9.6%, respectively. The superior performance in early OD, which involves much smaller objects, highlights the effectiveness of SPA in accurately selecting informative tokens, guided by multi-scale selection labels.

**Efficiency Analysis.** The GFLOPs reported in Table 1, Table 2, and Table 3 are computed over backbone, FPN and detection head with RGB input image at the resolution of $1280 \times 800$ for training stage. For a clearer comparison, we evaluate the throughput (i.e., FPS) only over the backbone network on a machine with an NVIDIA L40S GPU, as including other components would result in

Table 4: SPT also performs better for instance segmentation on COCO2017.

| Methods | AP | $AP_{50}$ | $AP_{75}$ | $AP_S$ | $AP_M$ | $AP_L$ |
|---|---|---|---|---|---|---|
| DynamicSwin-T (Rao et al., 2023) | 38.8 | 61.8 | 41.7 | 20.1 | 40.8 | 57.7 |
| **SPT-T** (ours) | **39.3** | **62.4** | **42.1** | **20.4** | **41.5** | **58.1** |
| DynamicSwin-S (Rao et al., 2023) | 39.6 | 63.1 | 42.8 | 21.0 | 42.4 | 59.2 |
| **SPT-S** (ours) | **40.9** | **64.6** | **44.0** | **21.8** | **43.9** | **60.1** |

values that are too small. From these tables, we obtain the following foundings: 1) SPT consistently outperforms all baselines in terms of efficiency, achieving a reduction in backbone computation cost by 10.9% to 24.9%. 2) The efficiency improvement is more pronounced when datasets contain smaller objects. For instance, the computation reduction of SPT-T on BDD-S is 20.8%, compared to 16.8% on the full BDD100K dataset. This aligns with expectations, as our approach selects only object-containing tokens for computation. 3) To achieve better performance, SPA is applied only at the last two stages, with four additional blocks introduced at the third stage. However, due to downsampling at the third stage, the number of tokens is reduced by a factor of $16\times$, leaving the early stages to dominate computation costs. Despite SPA selecting only 22% of tokens for computation (discarding 78%), the overall efficiency improvement is limited to 16.8%.

## 4.3 SPT FOR OTHER COMPUTER VISION TASKS

In addition to the object detection task, we also evaluated our SPT on other tasks, including instance segmentation and multi-label classification.

**Instance Segmentation.** As shown in Table 4, both tiny and small versions of SPT outperform baseline models (Rao et al., 2023). SPT-S achieves a more substantial improvement, increasing AP from 39.6 to 40.9 on COCO2017.

Table 5: The SPA blocks reduce the computation with a low number of selected tokens (i.e., select ratio) and achieve better performance in multi-label classification on PASCAL VOC 2012.

| Dataset | Methods | Mean Select Ratio(%) | mAP |
|---|---|---|---|
| **PASCAL VOC** | DynamicSwin-T (Rao et al., 2023) | - | 44.12 |
| | **SPT-T** (ours) | **29.6** | **44.60** |

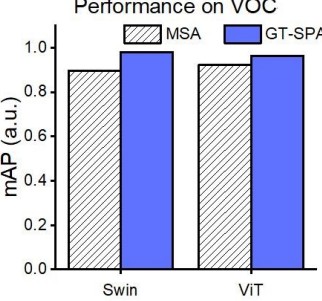 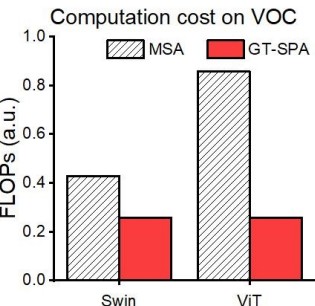

Figure 4: Under ground truth supervision, attending to only informative tokens can achieve better performance and efficiency.

**Multi-Label Classification.** For multi-label classification on PASCAL VOC 2012, SPA improves performance to 44.6, outperforming DynamicSwin (Rao et al., 2023), as shown in Table 5. Additionally, we present the mean select ratios for SPA blocks. Overall, SPT reduces GFLOPs by 10.2%.

Table 6: Starting to replace the Swin blocks from the third stage performs the best, as early-stage selection leads to information loss. $\mathcal{L}_{select}$ significantly reduces the number of selected tokens for computation, with lower select ratios. AP and FLOPs in this table are evaluated with $\mathcal{L}_{select}$.

| Dataset | SPA Blocks | | | | Select Ratios of stages (w/o $\mathcal{L}_{select}$) | | | | Select Ratios of stages (w/ $\mathcal{L}_{select}$) | | | | AP | AP$_{50}$ | AP$_{75}$ |
|---|---|---|---|---|---|---|---|---|---|---|---|---|---|---|---|
| | $1_{st}$ | $2_{nd}$ | $3_{rd}$ | $4_{th}$ | $1_{st}$ | $2_{nd}$ | $3_{rd}$ | $4_{th}$ | $1_{st}$ | $2_{nd}$ | $3_{rd}$ | $4_{th}$ | | | |
| BDD100K | ✗ | ✗ | ✗ | ✓ | ✗ | ✗ | ✗ | 36.5 | ✗ | ✗ | ✗ | 25.02 | 21.9 | 32.7 | 24.2 |
| | ✗ | ✗ | ✓ | ✓ | ✗ | ✗ | 85.02 | 52.41 | ✗ | ✗ | 23.21 | 25.01 | **22.6** | **33.1** | **24.6** |
| | ✗ | ✓ | ✓ | ✓ | ✗ | 99.24 | 82.15 | 71.45 | ✗ | 20.18 | 22.56 | 25.0 | 20.5 | 31.3 | 22.3 |
| | ✓ | ✓ | ✓ | ✓ | 83.49 | 99.12 | 88.64 | 79.86 | 13.42 | 20.78 | 22.36 | 25.0 | 18.3 | 29.4 | 20.6 |

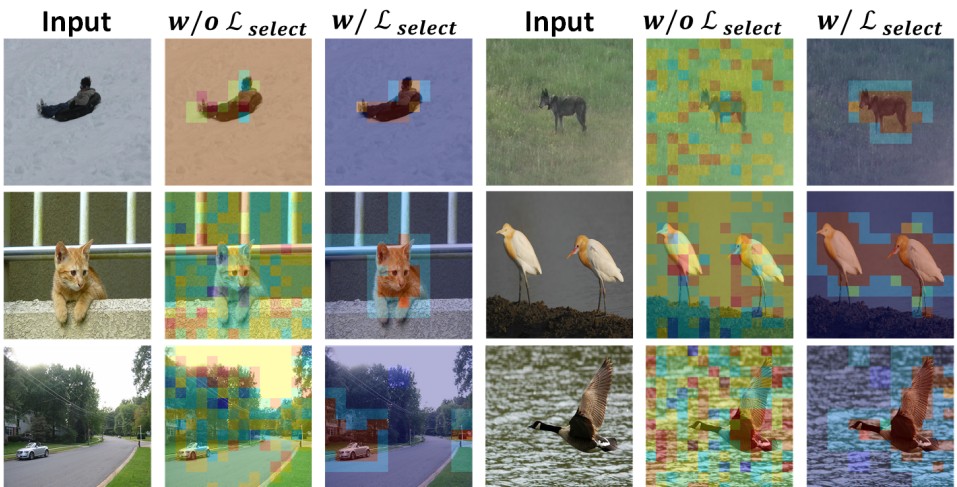

Figure 5: We overlay the summation of the selection masks generated by all SPA blocks on the original image. Warm color denotes high frequency of selection while cold color means be pruned before the attention computation. With multi-scale supervision, the selection process is more accurate.

## 4.4 ABLATION STUDY

**The Effect of Token Selection for Attention.** To illustrate the effectiveness of informative token selection, we designed experiments where all informative tokens were selected based on ground truth selection. As illustrated in Fig. 4, for both plain ViT and window-based attention mechanisms (i.e., Swin), selecting tokens to disregard background information improves both accuracy and efficiency on PASCAL VOC 2012 dataset.

**Specific Design for Selection.** Even though we know that token selection works, a critical challenge is how to correctly select these informative tokens without ground truth labels. As discussed earlier, previous methods (e.g., SparseViT) commonly adopt uniform token selection, applying a fixed ratio for all images in each batch. However, the results in Table 7 demonstrate that our SPA with dynamic selection performs better. Additionally, Fig. 5 provides visual comparisons to illustrate the effectiveness of our proposed multi-scale select label.

Table 7: For uniform sparse attention, we adopt the top-50 technique as SparseViT (Row 1). SPA block performs better. $\mathcal{L}_{select}$ further improves the performance.

| Dataset | SPA | $\mathcal{L}_{select}$ | Mean Select Ratio(%) | mAP |
|---|---|---|---|---|
| PASCAL VOC | ✗ | ✗ | 50 | 44.42 |
| | ✓ | ✗ | 59.77 | 44.49 |
| | ✓ | ✓ | **29.60** | **44.60** |

**Number of SPA blocks.** Table 6 explore the optimal number of SPA blocks in SPT. The results match with the findings in Xia et al. (2022). Starting from the third stage yields the best performance. Early-stage selection leads to information loss, resulting in worse performance.

## 5 CONCLUSION

In this paper, we analyze the current issues with sparse attention mechanisms and propose a novel Select and Pack (SPA) mechanism to address these challenges for both efficiency and performance. SPA focuses attention computations solely on informative tokens using a supervised gating block in Vision Transformers and packs the selected tokens for parallelized GPU batch training and inference. Integrated into the Swin Transformer's hierarchical architecture, SPA forms the efficient Select and Pack Transformer (SPT), which works as image backbone network for various computer vision tasks and generates multi-scale representations. Extensive experiments across three datasets and a range of vision tasks validate the effectiveness of SPT.

Despite these advantages, SPA relies on spatial supervision (e.g., bounding boxes or masks) to guide token selection; while this is naturally available in detection and instance segmentation, its benefits may diminish in tasks lacking structured spatial labels or in highly dense-label settings. Exploring self-supervised or task-adaptive selection strategies represents a promising direction for future work.

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

## APPENDIX

## A    GATE MECHANISM IN THE SELECT-AND-PACK MODULE

This section provides additional details on the gating mechanism used in the Select-and-Pack (SnP) module, complementing the description in Section 3.2. Similar to FLASH attention (Hua et al., 2022) which proposes a transformer with linear time complexity, utilizing Gated Linear Units (GLUs) (Shazeer, 2020), the gate is a lightweight, single-layer linear projection applied to each token embedding to produce a scalar importance score. Formally, given token feature $x_i$, the gate computes:

$$s_i = \sigma(W_g x_i + b_g) \tag{11}$$

These scores $s_i$ represent token informativeness and are supervised using the multi-scale selection labels described in Section 3.2.

To enable end-to-end training while sampling discrete tokens, we use the Gumbel-Softmax straight-through estimator. The sampling probability for each token is:

$$\tilde{s}_i = \text{Gumbel-Softmax}(\boldsymbol{s}_i, \tau) \tag{12}$$

where $\tau$ is the temperature parameter.

This allows gradients to flow through the sampling operation during training.

## B    MORE RESULTS AND ANALYSIS

### B.1    SELECT LOSS

We experimented with weighted cross-entropy to emphasize the selection of positive tokens. This approach indeed increased the number of selected tokens. However, it also included more background tokens, which resulted in worse performance. For example, when the weight was set to 5 in our experiment on BDD-S, the selection ratio increased from 33% to 71%, but the final mAP dropped to 5.3.

### B.2    SPT LIGHTWEIGHT VARIANTS

We actually offer two variants of the proposed SPT. The version shown in the main text (SPT-T/16) emphasizes higher accuracy and thus introduces slightly more transformer blocks, resulting in a modest increase in parameter count.

To provide a fairer comparison in terms of model size and efficiency, we also implemented a lightweight variant (SPT-T/12) that maintains a similar parameter count to DynamicSwin-T while achieving better performance and higher FPS. Table 8 provides a detailed comparison on the BDD100K dataset.

Table 8: SPT-T/16 achieves the best performance with slightly higher computational overhead compared to SPT-T/12; however, its FPS remains lower than that of DynamicSwin-T (Rao et al., 2023).

| Model | AP | #Params | FPS |
|---|---|---|---|
| DynamicSwin-T (Rao et al., 2023) | 22.0 | 48 | 46 |
| SPT-T/12 | 22.3 | 48 | 62 |
| SPT-T/16 (Main text) | 22.6 | 55 | 58 |

### B.3 Object-Token Retention Analysis

We analyzed the proportion of object-containing tokens retained by SPA after selection. An object-containing token is defined as any token whose receptive field overlaps with a ground-truth object mask by $\geq 1$ pixel.

Results show that 92.3% of ground-truth object tokens are retained, with the minor loss (7.7%) confined mostly to non-critical object boundaries. Furthermore, the density of object-containing tokens improves drastically from 11.2% to 49% after screening.

### B.4 Performance–Efficiency Trade-off Analysis

Unlike traditional sparse attention methods that require predefined pruning ratios, SPA learns an input-adaptive selection policy. This allows the model to retain more tokens for complex scenes and prune aggressively when the background is simple.

To validate this design, we compare SPT with fixed pruning baselines. We enforce a fixed percentage of tokens to be kept at Stages 3–4 and evaluate performance.

Fig. 6 shows that Dynamic SPA achieves the highest accuracy (22.6 AP) even while pruning more tokens (74%) on average than fixed 50% pruning. Fixed heavy pruning (80%) causes a noticeable accuracy drop. This demonstrates that input-adaptive token selection is crucial: scenes with dense traffic naturally require more tokens to be preserved, while sparse scenes allow aggressive pruning.

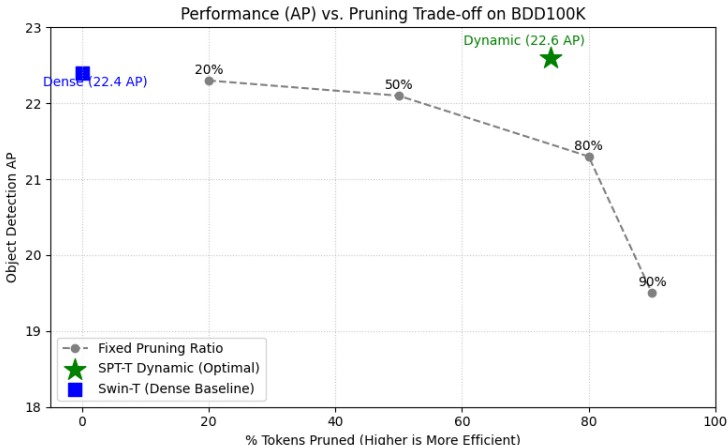

Figure 6: Performance vs. Pruning Level on BDD100K.

## C Limitations

Currently, the proposed method needs segmentation masks or bounding boxes to create select labels for supervision. Although our main target task is object detection, some vision tasks may lack this kind of labels. While some models can be used to generate these labels, like SAM (segment-anything), the generation process may introduce some errors.

## D Impact Statement

This paper presents work whose goal is to advance the field of Machine Learning. There are many potential societal consequences of our work, none which we feel must be specifically highlighted here.

## E  SOURCE CODE

We provide the anonymized source code in a zip file.

