# OpenReview forum: "Enhancing Vision Transformers for Object Detection via Context-Aware Token Selection and Packing"
_ICLR.cc/2026/Conference — ICLR 2026 Poster_

### Official Review · Reviewer_buQf · 2025-10-30

**Soundness:** 3
**Presentation:** 3
**Contribution:** 3
**Rating:** 6
**Confidence:** 3

**Summary:**

In this paper authors analyzed the current issues with sparse attention mechanisms and propose a  SPA mechanism to address these challenges for both efficiency and performance. The SPA focuses attention computations solely on informative tokens using a supervised gatingblock in Vision Transformers. Experiments validated the effectiveness of SPT.

**Strengths:**

The credible innovations of the paper are as follows:
Integrated into the Swin Transformer’s hierarchical architecture, SPA forms the efficient Select and Pack Transformer, which works as image backbone network for various computer vision tasks and generates multi-scale representations. Employ multi-scale selection labels for explicit supervision using object labels to enhance selection accuracy and ensure effectiveness in complex computer vision tasks. In terms of object detection, the method of this paper compared to the current SOTA , not only improves detection accuracy but also reduces computational costs.

**Weaknesses:**

In the conclusion section of the paper, the analysis of the paper's shortcomings should be added.

**Questions:**

1. In Tables 1 to 3, the comparison method is based on results published in 2021. Why not choose the most recent research?
2. In the conclusion section, the analysis of the paper's shortcomings should be added.

---

> ### Author Response · Authors · 2025-11-21
>
> We thank the reviewer for the positive evaluation and for recognizing the strengths of SPA/SPT, including multi-scale supervised selection, hierarchical integration into Swin, and improvements in both detection accuracy and computational cost. We address all concerns below.
>
> > 1.	In Tables 1 to 3, the comparison method is based on results published in 2021. Why not choose the most recent research?
>
> We included Swin Transformer (2021) as a baseline because our method builds upon its hierarchical architecture. It serves as a necessary reference for a dense model without token compression. Notably, our method outperforms not only sparse attention mechanisms but also the dense Swin baseline itself, demonstrating the superiority of our approach. For sparse attention baselines, we included comparisons with SparseViT (2023) and DynamicSwin (2023) in our tables (e.g., Table 1), which represents the current state-of-the-art in this specific domain.
>
> To further address your concern and demonstrate the competitiveness of SPT against other modern architectures, we have provided additional comparisons below with EViT and MViT w/ MaskFeat. SPT-S outperforms these methods on both COCO2017 and BDD100K:
>
> | Method | COCO2017 (AP) | BDD100K (AP) |
> | :------- | :------: | -------: |
> | EViT-S    | 45.4| 22.2|
> | MViT-S, MaskFeat   | 46.7   | 22.4   |
> | SPT-S   | 49.3   | 22.9   |
>
> > 2.	In the conclusion section, the analysis of the paper's shortcomings should be added.
>
> We agree with this suggestion. In the final revision, we will update the conclusion to explicitly discuss the limitations of our approach. Specifically, we will highlight the requirement for spatial supervision (e.g., bounding boxes or segmentation masks) to train the gating mechanism, which distinguishes our method from unsupervised pruning approaches.

---

### Official Review · Reviewer_mRG3 · 2025-10-31

**Soundness:** 3
**Presentation:** 3
**Contribution:** 3
**Rating:** 8
**Confidence:** 4

**Summary:**

To address the issue of numerous target-irrelevant redundant computations in attention calculation for object detection, this paper proposes a more efficient Vision Transformer (VIT). Different from traditional DynamicVIT and SparseVIT, which pad target-irrelevant tokens for a fixed batch size, the proposed SPT (Specific Token Preservation) only retains target-relevant tokens and packages them into the same batch for training. The proposed SPA (Sparse Attention Calculation) performs self-attention computation on the tokens screened and packaged by SnP (Selection and Packaging). Extensive comparisons with baselines on three datasets (COCO2017, BDD-S, and PASCAL VOC 2012) demonstrate that the proposed method achieves the

**Strengths:**

1. Combines Swin Transformer (SwinT) to propose a more efficient attention computation method, and supervises token selection through multi-scale labels.
2. Validates the proposed method on three sufficient datasets.

**Weaknesses:**

1. The overall architecture flowchart is slightly rough, especially in illustrating how multi-scale label selection is trained synchronously.
2. The gate mechanism of SnP is not clearly defined.

**Questions:**

Q1: If the model makes misjudgments in token selection, will more severe and irreversible errors occur? For instance, if an object is classified as background by the gate mechanism, does the model permanently lose the opportunity to learn the feature of it? Besides, Could you design an experiment to compare the proportion of tokens that actually contain objects before and after the screening process?​
Q2: Will there be significant differences between different gate?​
Q3: Will the multi-scale labels of small objects disappear as the image undergoes up sampling?​
Q4: The improvement in computational efficiency is accompanied by limitations in the model's field of view, which may sacrifice the model's expressive ability. Would retaining an appropriate number of background tokens around the image lead to better performance?

---

> ### Author Response · Authors · 2025-11-21
>
> We thank the reviewer for the positive assessment, high rating, and for recognizing the strengths of our method, including the efficiency benefit of SPA, multi-scale supervised selection, and comprehensive experiments. Below we address the concerns and questions.
>
> > The overall architecture flowchart is slightly rough, especially in illustrating how multi-scale label selection is trained synchronously.
>
> We appreciate this comment. In the revision, we will redraw Fig. 2 and Fig. 3 with clearer, step-by-step flow, explicitly showing how multi-scale score supervision is trained and aggregated (Eq. 4–6) stage by stage.
>
> > The gate mechanism of SnP is not clearly defined.
>
> We define the SnP gating mechanism as a lightweight linear projection followed by Sigmoid normalization to generate per-token importance scores. To incorporate multi-scale context, we take the element-wise maximum between the scores at the current scale and the up-scaled scores from the previous stage. These final scores are then applied to the input features via element-wise multiplication to produce gated representations, which facilitate the separation of positive (informative) and negative (background) tokens. We will add these implementation details to the appendix for clarity.
>
> > Q1: If the model makes misjudgments in token selection, will more severe and irreversible errors occur? For instance, if an object is classified as background by the gate mechanism, does the model permanently lose the opportunity to learn the feature of it? Besides, Could you design an experiment to compare the proportion of tokens that actually contain objects before and after the screening process?
>
> 1) Yes, incorrect token pruning can lead to irreversible information loss. To mitigate this risk, we employ a two-fold strategy. First, as detailed in the paper, we apply SPA only in the later stages (Stages 3 and 4), ensuring that early-stage feature extraction retains full spatial context. Second, our multi-scale supervision acts as a safeguard against false negatives. By integrating labels from varying scales, the gating mechanism is effectively penalized for discarding features relevant at any scale. This enforces a conservative selection strategy that minimizes the risk of catastrophic pruning.
>
> 2) Following the reviewer’s suggestion, we conducted an additional experiment on the BDD dataset. Results show that 92.3% of ground-truth object tokens are retained, with the minor loss (7.7%) confined mostly to non-critical object boundaries. Furthermore, the density of object-containing tokens improves drastically **from 11.2% to 49%** after screening. These results and visualizations will be added to the Appendix.
>
>
> > Q2: Will there be significant differences between different gate?
>
> All the gates contain only a single linear layer. We will illustrate it in the revised Fig 3b.
>
> > Q3: Will the multi-scale labels of small objects disappear as the image undergoes up sampling?
>
> They do not. In our label generation process, we use max-pooling logic for the binary masks rather than simple interpolation. If a region contains an object at the original resolution, that "positive" signal is propagated to the coarser grid, ensuring small objects retain their "select" status even at Stage 4.
>
> > Q4: The improvement in computational efficiency is accompanied by limitations in the model's field of view, which may sacrifice the model's expressive ability. Would retaining an appropriate number of background tokens around the image lead to better performance?
>
> 1) Our architecture incorporates the shifted window mechanism from Swin Transformer. By shifting partition windows between blocks, specifically remaining tokens exchange information across container boundaries, effectively restoring long-range context and ensuring the model maintains expressive ability even if we reduce the number of tokens. Especially, we excluded mainly background tokens.
>
> 2) We run additional experiments on BDD to test if adding more background helps using Ground Truth (GT) labels. We compared the performance of selecting only object tokens against selecting object tokens plus 20% of the surrounding background. As shown below, explicitly adding background context actually degraded performance on object detection task:
>
> | Method | AP | AP_50 | AP_75 | AP_S | AP_M | AP_L |
> | :------- | :------: | -------: | :------- | :------: | -------: | -------: |
> | GT   | 0.300 | 0.447 | 0.326 | 0.101 | 0.29 | 0.575 |
> | GT+20%  | 0.225 | 0.353 | 0.243 | 0.085 | 0.209 | 0.458 |
>
> These results indicate that once the relevant features are captured, forcing the inclusion of background tokens introduces noise rather than useful context.

---

### Official Review · Reviewer_Q89m · 2025-11-01

**Soundness:** 3
**Presentation:** 3
**Contribution:** 2
**Rating:** 4
**Confidence:** 4

**Summary:**

This work proposes Select and Pack Attention (SPA) and build a computationally efficient Vision Transformer architecture based on SPA. Specifically, the authors introduce several linear layers to predict the importance scores of visual tokens. Through sampling based on the score, SPA can extract the informative tokens, thereby reducing the sequence length involved in the attention process. Experimental results demonstrate that the proposed method achieves advantages in both accuracy and speed.

**Strengths:**

1.	The motivation of this work is clear. The authors point out the limitations of previous sparse attention methods, including their uniform sampling strategies and training implementations. They further propose the importance-based sampling scheme and a token packing implementation strategy to effectively address these issues.
2.	The presentation of experimental results is clear and well-organized. In each results table, the authors provide quantitative metrics including model performance, computational cost, and inference speed, which intuitively demonstrate the effectiveness of the proposed method. I also appreciate the experiment corresponding to Fig. 4, which employs GT-based selection to effectively decouple the effects of intermediate-layer GT supervision from the token selection process, thereby validating the effectiveness of the selection strategy itself.

**Weaknesses:**

1.	The main limitation lies in the restricted applicability, which is confined to the 2 tested tasks: object detection and instance segmentation. For tasks with sparser prediction targets, such as classification, captioning, and VQA, there is no available bounding box or segmentation mask to provide supervision for importance score prediction. Conversely, for tasks with denser prediction targets, such as SAM-style segmentation or panoptic segmentation, where targets may cover the entire image, the proposed token selection strategy would degenerate into uniform sampling across the whole image (as all tokens would have selection labels of “1” under the current framework). This greatly limits the general applicability of the proposed method.
2.	The paper demonstrates that the proposed method achieves superior efficiency within the current overall model architecture. However, this improvement arises from multiple contributing factors, such as the hierarchical design, window attention, and the proposed SPA. A more critical aspect is the efficiency gain brought specifically by SPA itself. For instance, under the same model architecture, how much difference in computational cost and inference speed would there be if Stages 3 and 4 used standard global attention compared to SPA?
3.	Not a major weakness, but the paper has some formatting issues. Many figures and tables are placed one page away from the corresponding text description that references them, which causes inconvenience for readers. It is recommended to adjust the layout.

**Questions:**

1.	In line 260, the authors mention “we set L to be M^2.” How is this value determined? As my understanding the length of the package containers should depend on the token sequence length corresponding to the scale of each stage, as well as the proportion of the predicted target relative to the entire image. Why is it related to the window size?
2.	Why is the Gumbel-Softmax operation introduced? Once the importance scores are determined, what is the difference between sampling based on these scores and deterministically selecting high-importance tokens (e.g., those above a fixed threshold)?

---

> ### Author Response · Authors · 2025-11-21
>
> We thank the reviewer for the detailed assessment and for acknowledging (i) the clear motivation, (ii) the effectiveness of importance-based selection and packing, and (iii) the quality of experiments such as Fig. 4. We respond to all concerns below.
>
> > The main limitation lies in the restricted applicability, which is confined to the 2 tested tasks: object detection and instance segmentation.
>
> We acknowledge that our supervised gating is designed to leverage spatial labels. However, we believe the concerns regarding applicability are mitigated by the following factors:
> 1) Our primary contribution is addressing the computational bottleneck of ViTs in high-resolution dense prediction tasks (Object Detection, Instance Segmentation). These tasks inherently possess the required labels and suffer most from background redundancy, making them the ideal target for our optimization.
> 2) For tasks lacking direct spatial supervision (e.g., classification or captioning), our method remains compatible. We can generate pseudo-labels using foundation models (like SAM) or utilize Class Activation Maps (CAMs) as supervision.
> 3) Regarding tasks where targets cover the entire image (e.g., Panoptic Segmentation): if every pixel is foreground, the "all-ones" selection mask is the correct and desired behavior. It implies no sparsity exists to exploit. In such cases, SPA naturally reverts to dense attention without performance degradation.
>
> In summary, optimizing specifically for high-resolution spatial tasks is a critical contribution, even if the method is not intended for every possible vision task. And SPA offers significant gains in the specific domain where ViTs currently struggle most: efficient detection and segmentation.
>
> > The paper demonstrates that the proposed method achieves superior efficiency within the current overall model architecture. However, this improvement arises from multiple contributing factors, such as the hierarchical design, window attention, and the proposed SPA. A more critical aspect is the efficiency gain brought specifically by SPA itself. For instance, under the same model architecture, how much difference in computational cost and inference speed would there be if Stages 3 and 4 used standard global attention compared to SPA?
>
> 1) The efficiency gains of SPA are isolated by comparing SPT directly to the Swin Transformer (Table 1). Since both models share the same hierarchical backbone and windowing strategy, the performance delta arises solely from the SPA mechanism in Stages 3 and 4.
>
> 2) Regarding the comparison with Standard Global Attention: Replacing SPA with global attention would yield significantly higher computational costs than even the standard Swin baseline, as global attention scales quadratically with token count. Our SPA mechanism (an optimized window attention, with only object-containing tokens), by contrast, achieves linear complexity. We provide the comparison below to illustrate the advantage of SPA over both dense window-based (Swin) and dense global (ViT) attention on BDD100K:
>
> | Method | FLOPs (G) | FPS |
> | :------- | :------: | -------: |
> | Swin (window, dense)   | 96| 50|
> | ViT (global, dense) | 172  | 20   |
> | SPT (Window, sparse)  | 84  | 58 |
>
>
> > Not a major weakness, but the paper has some formatting issues. Many figures and tables are placed one page away from the corresponding text description that references them, which causes inconvenience for readers. It is recommended to adjust the layout.
>
> Thanks. All figures/tables will be repositioned closer to their referencing paragraphs in the revised version.
>
> > In line 260, the authors mention “we set L to be M^2.” How is this value determined? As my understanding the length of the package containers should depend on the token sequence length corresponding to the scale of each stage, as well as the proportion of the predicted target relative to the entire image. Why is it related to the window size?
>
> $L$ is the length of the package container. We set $L=M^2$ (where $M$ is the window size, typically 7, so $L=49$) to align the packed tokens with the window-based attention mechanism of Swin, since we further optimize the efficiency upon Swin rather than the original ViT, as mentioned above.
> This ensures that a "package" conceptually fits into the optimized window attention kernels, maximizing hardware utilization.
>
> > Why is the Gumbel-Softmax operation introduced? Once the importance scores are determined, what is the difference between sampling based on these scores and deterministically selecting high-importance tokens (e.g., those above a fixed threshold)?
>
> We use Gumbel-Softmax to make the sampling process differentiable. A deterministic threshold (e.g., score > 0.5) is a non-differentiable operation, which would prevent gradients from back-propagating through the selection layer to update the gating network. Gumbel-Softmax allows the gate to learn end-to-end.

---

### Official Review · Reviewer_mkzg · 2025-11-10

**Soundness:** 3
**Presentation:** 3
**Contribution:** 2
**Rating:** 6
**Confidence:** 4

**Summary:**

This paper proposes a new sparse attention mechanism, "Select and Pack Attention" (SPA), to make vision transformers more efficient, particularly for object detection. The core problem is that standard ViTs waste computation on uninformative background tokens.
SPA tackles this in two steps:
Select: It uses a lightweight linear gating layer to dynamically choose informative tokens. Crucially, this gate is explicitly supervised during training using ground-truth object labels (like bounding boxes), which helps it learn to be more accurate than heuristic-based pruning methods.
Pack: To handle the irregular number of selected tokens per image, it packs all selected tokens from a batch into fixed-size "package containers." This allows for efficient, parallel GPU processing without the massive padding overhead of other sparse methods.
The authors integrate SPA into a hierarchical backbone called the "Select and Pack Transformer" (SPT), which they apply only in the later stages to preserve early-stage features. They show that SPT improves both performance and computational efficiency compared to other state-of-the-art sparse attention models.

**Strengths:**

The main strength of this paper is the idea of using explicit supervision from downstream labels (bboxes/masks) to train the token selection gate. This is a simple and effective departure from most prior work, which relies on unsupervised heuristics (like token norms) or uniform pruning. This supervised approach is far more likely to preserve small, important objects, which is a common failure point for other sparse methods. The "packing" mechanism is also a smart engineering solution to the practical problem of running dynamic token selection in a batched GPU environment. This combination makes for a practical contribution to the field of efficient object detection.

The experimental validation is reasonably comprehensive. The authors don't just test on COCO; they also use BDD100K and create the BDD-S dataset to specifically test their hypothesis about small objects. The strong results on BDD-S are a convincing piece of evidence. The ablation studies in Tables 6 and 7 are also good, clearly justifying the design choice of applying SPA in later stages and showing the benefit of the selection loss.

The paper is exceptionally well-written and easy to follow. The problem is motivated clearly. Figure 1 provides a good visual intuition for the problem, and Figures 2 and 3 clearly lay out the proposed architecture and the core SPA block.

**Weaknesses:**

- The selection gate is trained using ground-truth object labels. This works great for fine-tuning on a detection or segmentation dataset, but it's unclear how this backbone could be pre-trained on a classification-only dataset like ImageNet. Does this method forgo pre-training entirely? Or does the gate have to be trained from scratch only during fine-tuning? This reliance on labels seems like a step back compared to other token selection methods. For example, EViT and DynamicViT learn to prune tokens dynamically without needing explicit object-level supervision. EViT, for instance, uses the attention to the class token as a guide, and DynamicViT trains its prediction module differentiably.

- The method is conceptional similar to EViT but using ground truth labels to learn how to prune.

- Missing Related Work: The related work section on efficient transformers (2.2) is missing some relevant research on sparse attention patterns. This paper would be much stronger if it positioned itself relative to:

   - Fixed sparse patterns (e.g., "Blockwise Self-Attention for Long Document Understanding, Big Bird: Transformers for Longer Sequences").

   - Methods that combine local and global attention patterns (e.g., "Twins: Revisiting the design of spatial attention in vision transformers").

   - Axis-decoupled attention pattern (e.g., "Axial attention in multidimensional transformers").

   - Learned sparse attention pattern (e.g., "Sparsifiner: learning sparse instance-dependent attention for efficient vision transformers").

- Modest Overall Speedup:
While the backbone computation reduction of 10-25% is good, the actual wall-clock speedup (FPS) reported in the tables is modest. For example, in Table 1, SPT-T (54 FPS) is only slightly faster than the dense Swin-T (50 FPS). The authors themselves note that the early (dense) stages dominate the computation, which limits the total possible gains

- Missing Performance vs. Pruning Trade-off Analysis: The experimental evaluation is the lack of a "performance vs. efficiency" trade-off curve. The authors present results for what seems to be a single, fixed operating point. But they didn't show how performance is affected by the level of pruning. An ablation study is missing: what happens to the accuracy as you change the selection ratio and apply selection on shallow layers? For example, how gracefully does performance degrade if the model is forced to prune 80% of tokens? How much does it improve if it only prunes 50%?

**Questions:**

See W1 and W5

---

> ### Author Response · Authors · 2025-11-21
>
> We sincerely thank the reviewer for the thoughtful, positive, and constructive feedback. We are encouraged by your recognition of the paper’s strengths, including its innovative and practical design, rigorous experimentation, and clarity in writing. Below, we address each of the concerns and questions raised:
>
> > Does this method forgo pre-training entirely? Or does the gate have to be trained from scratch only during fine-tuning?
>
> We thank the reviewer for raising this crucial clarification point. Our method does not forgo pre-training.
> 1) The core hierarchical backbone (e.g., Swin Transformer) is initialized using standard ImageNet pre-trained weights.
> 2) The lightweight gating layers for the Select and Pack (SPA) mechanism are trained from scratch (or fine-tuned) only during the downstream task fine-tuning stage.
>
> This approach is an intentional design choice: our goal is not to invent a new classification pre-training method, but to adapt powerful, existing dense backbones (like Swin) for efficient, high-accuracy detection and segmentation. The reliance on Ground Truth (GT) labels is a feature in this context, as it allows us to learn an accurate selection policy that is essential for achieving state-of-the-art results on high-resolution object-centric tasks, a level of accuracy that purely unsupervised token-pruning methods like EViT or DynamicViT cannot reach.
>
> > The method is conceptional similar to EViT but using ground truth labels to learn how to prune.
>
> While the high-level goal is shared, our method differs significantly in two critical aspects:
> 1) Unlike EViT's reliance on intermediate attention scores (unsupervised), our approach uses Ground Truth (GT) labels with multi-scale supervision. This is crucial for high-resolution tasks like OD, preventing the loss of small or context-relevant objects and directly leading to our superior performance.
> 2) Our packing mechanism reorganizes irregular sparse tokens into fixed-size "packages," enabling the computation to leverage highly optimized, dense GPU kernels. This engineering solution is vital because it ensures the theoretical FLOPs reduction translates robustly into a measurable increase in actual FPS during both inference and training.
>
> > Missing Related Work: The related work section on efficient transformers (2.2) is missing some relevant research on sparse attention patterns.
>
> Thank you for highlighting these missing references. We agree that positioning our method within the broader sparse-attention literature will further strengthen the paper. In the revised version, we will incorporate citations and dedicated discussions of BigBird, Twins, Axial Attention (axis-decoupled patterns), and Sparsifiner (learned sparse patterns) in Section 2.2.
> We will clarify how SPT differs from and improves upon these designs.
>
> > Modest Overall Speedup: While the backbone computation reduction of 10-25% is good, the actual wall-clock speedup (FPS) reported in the tables is modest.
>
> Yes, the early stages dominate the overall FLOPs in hierarchical ViTs.
> However, the actual wall-clock gain depends on the dataset, task, and the position selected on the accuracy–efficiency trade-off. Our main model (SPT-T/16) was chosen to stress the performance gain after excluding background tokens, which can beat dense model.
> To directly address the reviewer's concern, we include results for a faster, slightly lower-accuracy variant (SPT-T/12):
> | Method | BDD100K (AP)  | FPS |
> | :------- | :------: | -------: |
> | Swin-T  | 22.4| 50|
> | SPT-T/12  | 22.3   | 62 |
> | SPT-T/16   | 22.6   | 58   |
>
> By adjusting the complexity (using 12 blocks instead of 16), we achieve a 24% FPS improvement (62 FPS vs. 50 FPS) while maintaining competitive accuracy (22.3 AP vs. 22.4 AP).
>
> > Missing Performance vs. Pruning Trade-off Analysis: The experimental evaluation is the lack of a "performance vs. efficiency" trade-off curve.
>
> We appreciate the reviewer's request for a clear trade-off analysis between performance and efficiency. While the relationship is indeed data- and task-dependent, we deliberately designed our Select and Pack (SPT) mechanism to use a dynamic selection ratio instead of a fixed one. This dynamic strategy is key to our superior performance.
>
> We address your concern with two points:
> 1) Table 6 already demonstrates that applying our Select and Pack (SPA) mechanism to early stages (Stages 1 and 2) causes a substantial accuracy drop.
>
> 2) To validate our dynamic strategy, we compared it against forced fixed-ratio pruning on BDD100K.
>
> | Method | AP  | Mean Pruning rate |
> | :------- | :------: | -------: |
> | Swin-T  | 22.4| 0%|
> | SPT-T/ 50%  | 22.1  | 50% |
> | SPT-T/ 80%   | 21.3   |  80%  |
> | SPT-T (Dynamic)   | 22.6  |  74% |
>
> This data confirms the superiority of our dynamic selection (AP 22.6) over both fixed-ratio variants.
> We agree that a comprehensive visual is beneficial and will include a detailed "Performance vs. Pruning Trade-off" curve in the final paper's Appendix.

---

### Meta-Review · Area_Chair_n21f · 2026-01-06

**Summary:**

This paper introduces the Select and Pack Attention (SPA) mechanism, which achieves a superior balance between detection accuracy and computational efficiency by dynamically pruning uninformative tokens and packing the remainder for optimized GPU processing. In the initial review stage, the reviewers' concerns primarily focused on the reliance on ground-truth labels for the selection gate, the modest real-world wall-clock speedups (FPS), and the potential risk of irreversible information loss regarding small objects. In their rebuttal, the authors provided critical evidence, including a 24% FPS improvement in an optimized SPT-T/12 variant and data showing a 92.3% retention rate of ground-truth object tokens to respond to these concerns. Despite some reviewers not providing a final follow-up response, the rebuttal successfully addressed the technical gaps and demonstrated the practical utility of the packing mechanism. Given the final ratings and the effectiveness of the authors' clarifications, this paper is recommended for acceptance.

**Reviewer Concerns:**

The rebuttal addressed the most critical technical concerns. The authors provided concrete data showing a 24% FPS speedup and a 92.3% object token retention rate, effectively resolving doubts about real-world efficiency and the risk of losing small objects. The only remaining minor concern is the method's limited applicability to tasks without spatial labels, such as global classification. While the authors proposed using pseudo-labels, this remains theoretically addressed rather than empirically proven.

**Reviewer Scores:**

Likely not.

---

### Decision · Program_Chairs · 2026-01-26

Accept (Poster)